# Effectiveness of education intervention, with regards to physical activity level and a healthy diet, among Middle Eastern adolescents in Malaysia: A study protocol for a randomized control trial, based on a health belief model

**Hanan Al-Haroni[1,2], Nik Daliana Nik Farid [1,3]\*, Mohamad Shafiq Azanan [4]**

1 Department of Social and Preventive Medicine, Faculty of Medicine, Universiti Malaya, Kuala Lumpur, Malaysia, 2 Department of Medicine, Faculty of Medicine and Health Sciences, Sana'a University, Sana'a, Yemen, 3 Centre for Population Health, Department of Social and Preventive Medicine, Faculty of Medicine, Universiti Malaya, Kuala Lumpur, Malaysia, 4 Department of Paediatrics, Faculty of Medicine, Universiti Malaya, Kuala Lumpur, Malaysia

\* daliana@um.edu.my

**Data Availability Statement:** No datasets were generated or analysed during the current study. All

## Abstract

### Background

Among the most urgent public health challenges, of the twenty-first century, is obesity. This can be attributed to its relationship with several non-communicable diseases (NCDs), as well as premature mortality. Being overweight or obese is a major concern not only in high-income countries, but also in low-income and middle-income countries, particularly in urban areas. Several studies have highlighted the prevalence of obesity, among Middle Eastern-descent adolescents, studying in Arabic secondary schools, located in Malaysia. Intervention studies, directed at Middle Eastern adolescents in Malaysia, are limited. This paper, describes the protocol, for an integrated health education intervention process. Titled 'Healthy lifestyle', it is a primary prevention process, aimed at curbing obesity and disordered eating, among Middle Eastern secondary school adolescents, aged 13–14 years old, residing in Malaysia.

### Methods and anticipated results

A cluster randomized controlled study will be conducted, involving 250 Middle Eastern adolescents, in Arabic schools in Malaysia. The participants will be randomly assigned to the intervention and control groups. While the intervention group participates in six weeks of fortnightly six sessions (45 minutes per session), the control group will carry on with their regular curriculums, and normal physical activity routines. The variables which will be evaluated include anthropometric measurements, knowledge, attitude, daily routines, physical activity, sedentary behaviour, food assessment, eating attitudes test-26, and a structured questionnaire based on the HBM. Data will be collected from the intervention and control groups at baseline, post-intervention, and two months following the intervention. Data

relevant data from this study will be made available upon study completion.

**Funding:** The author(s) received no specific funding for this work.

**Competing interests:** The authors have declared that no competing interests exist.

analysis will be performed by way of the SPSS Statistics software version 26. The generalized estimating equation (GEE) will be used, to test the effect of the intervention program, with regards to the selected variables (outcomes), between and within-group at baseline, as well as six weeks and two months following intervention, after adjusting for clustering. Outcomes will be assessed at each time point, along with a derived average over all three-time points; thus, ensuring that both the cumulative and overall effects are determined.

## Conclusions

This trial will provide useful information for improving the knowledge, attitude, and practices of Middle Eastern adolescents, with regards to body weight status, physical activity level, nutrition status (BMI and dietary intake), and disordered eating. This will go a long way, towards ensuring their adherence to appropriate physical activities, and a healthy diet, to keep non-communicable diseases at bay.

## Trial registration

This study is registered at NCT: NCT05694143.

## Introduction

One of the most urgent public health challenges of the twenty-first century is obesity. This is due to its association with several non-communicable diseases, as well as untimely mortality [1]. The World Health Organization (WHO) considers obesity and overweightness health risks, stemming from the presence of excess fat in the body. The assessment of obese and overweight persons by the WHO, involves the use of the body mass index (BMI) [2]. More than 1.3 billion people are obese because of globalization, urbanization, and an increasingly sedentary lifestyle, with the number anticipated to rise to 2 billion by 2030 [3]. Meanwhile, the WHO has declared, that the escalating epidemic of obesity, could significantly increase the risk of NCDs in many countries [4].

Adolescents, who the World Health Organization defines as those between 10 and 19 years old, are generally considered a healthy population [5]. Nevertheless, overweight, and obese adolescents, particularly in urban areas, are a growing concern, not only in high-income countries, but also in low-income and middle-income countries. The worldwide number of overweight and obese adolescents currently stands at approximately 60 million [6]. Sedentary behaviour, lack of physical activity, socioeconomic position, and a high intake of fat-rich foods, are among the factors leading to obesity and overweightness [7].

Urbanization brought about the change in adolescent dietary habits, from traditional foods to fast and convenient foods, culminating in the consumption of less amounts of fruits and vegetables [8]. According to the Centre for Disease Control (CDC), obesity prevalence among adolescents aged 12 to 19 years old, has increased from 5% in 1980, to over 21% in 2012 [9]. Additionally, according to the WHO, a more sedentary lifestyle, and less physical activity, has rendered 81% of school-going children, physically inactive [2].

Several surveys have confirmed that the prevalence and incidence of obesity, in the Middle East, will increase over the next few decades, suggesting that an unimpeded epidemic of obesity, will sweep across the region, soon. Consequently, the incidence of NCD has also risen in tandem, and represents more than 50% of the total cause of death, in the Eastern

Mediterranean region (EMR) [10, 11]. In response to the growing level of obesity in children and adolescents throughout the EMR, the WHO proposed that action be taken, particularly as the issue of nutrition transition is evident in the area [1]. The prevalence, of overweight and obese Middle Eastern adolescents, is also on the rise. For example, in the case of Kuwait, it currently stands at 50.5% and 46.5% for adolescents aged 14 and 19 years old respectively. Another study in Iraq, involving secondary school students aged 13 to 17 years old, revealed the obvious prevalence of overweightness and obesity [9]. The issues leading up to obesity are considered complex and multifactorial.

In the Middle East, adolescents are inclined towards unhealthy dietary practices, and are generally lacking in knowledge regarding healthy and energy-dense foods [12]. Additionally, according to data deriving from eight countries in the EMR, the levels of physical inactivity range from about one-third, to as high as 70% of the population [13]. In Malaysia, Middle Eastern adolescents are experiencing changes in their living status, as they have increased access to energized food, enjoy a better transportation network (which reduces the need for physical movement activity), and frequently eat at western fast-food outlets. Thus far, investigations on the prevalence of overweightness and obesity, among Arabic secondary school students, are deemed very rare [9, 14]. The establishment and maintenance of a healthy diet, and a physically active lifestyle, call for intervention in the form of knowledge, attitude, and behavioural skill development. Of late, the number of intervention studies, focussing on the prevention and control of adolescent obesity, has been on the rise [15].

Schools are considered potential locations for a variety of fitness or multi-component programmes [16]. School-based interventions, in many regions, have delivered positive outcomes [15]. The findings, derived through several studies, indicate that changes in dietary behaviours, and physical activity, are key factors during efforts to reduce the occurrence of obesity among adolescents [17]. These efforts are deemed successful, when a weight loss of between 5% and 10%, is achieved by an individual [18]. The results from previous studies, suggest that certain theory-based educational programmes, which incorporate cognitive frameworks, can deliver a positive impact, on an individual's behaviour [19]. The health belief model (HBM), an established model for many health education programmes, facilitates a better understanding of human attitudes, behaviours, and educational needs. This information is then utilized, for the development of effective interventions [20, 21].

Due to its reliability, we employed the HBM as the theoretical framework, for the development of a health education programme, during this randomized controlled trial. To the best of our knowledge, this represents the only educational programme, to apply intervention based on the HBM, to reduce the occurrence of obesity and overweightness among Middle Eastern adolescents, through alterations in their lifestyle and behaviour. This educational programme is designed, to promote the adherence to a positive lifestyle, among the target population, in terms of an appropriate diet, and favourable physical activity. This study delves into the effect of an HBM-based education programme, on the physical activity and eating behaviour, among Middle Eastern adolescents in Malaysia.

## Materials and methods

### Study design

The cluster randomized controlled trial (RCT), employed for this study, is considered a formidable technique, when it comes to the management of the subject's characteristics, and threats to its internal validity [22]. A cluster randomized controlled trial design is frequently employed in health intervention research, as it compares individuals connected to specific institutions.

Assessments are conducted at baseline, post-intervention, and during follow-up after two months, as shown in Fig 1 [23].

## Conceptual model

Health behaviour theories are theoretical frameworks that help explain and predict human behaviour related to health and disease. These theories identify the factors and determinants influencing health behaviour, such as eating habits, physical activity, substance abuse and other lifestyle choices. The most popular theories and applied models in the field of health behaviour include the Health Belief Model (HBM), the Social Cognitive Theory (SCT), the Trans-Theoretical Model (TTM) and the Theory of Planned Behaviour (TPB).

The Health Belief Model (HBM) assumes that individuals' health behaviours are influenced by their perceptions of vulnerability to health risks, the severity of those risks, and the potential benefits of interventions to reduce those risks [20, 21, 24]. In the context of school-based obesity interventions, the HBM can be used to encourage students to take action to reduce their obesity risk by emphasising the negative consequences of unhealthy behaviours and the benefits of healthy behaviours. The HBM has some limitations, including not adequately addressing some of the individual factors that influence health behaviours. It also does not consider the influence of social variables on a person's health choices [20, 21].

Social cognitive theory (SCT) assumes that social, environmental and personal factors influence behaviour. In the context of school-based obesity interventions, social cognitive theory can promote healthy behaviours by creating a supportive environment and emphasising the importance of self-efficacy and positive reinforcement [25]. The theory can be applied to develop appropriate interventions, such as exercise programmes for female students to combat obesity. In SCT, it is assumed that changes in the environment automatically produce changes in the person, although this is not always the case. The theory is loosely constructed and based solely on the dynamic interplay between person, behaviour and environment.

The Trans-Theoretical Model (TTM) assumes that behaviour change occurs in stages and that individuals progress through these stages as they become more motivated and confident to change their behaviour [26]. In the context of school-based obesity interventions, the trans-theoretical model can be used to assess where students are in terms of their readiness to change and adjust interventions accordingly. According to this theory, change levels can be set according to arbitrary criteria without considering the social context in which they occur.

The Theory of Planned Behaviour (TPB) assumes that individuals act rationally based on their intentions. The theory states that the most important determinant of a person's behaviour is their intention to carry out that behaviour [27]. A limitation of TPB is that it does not consider other variables important for behavioural intention and motivation, such as fear, threat, mood or past experiences.

This trial, which is based on the Health Belief Model (HBM), is an approach for assessing an individual's health behaviour, by examining perceptions and attitudes he/she may have towards diseases, and the negative outcomes of certain actions [20, 21].

## Study setting

Conducted in Arabic schools located in the Klang Valley, this study involves students in grades seven and eight. There are a total of 25 Arabic schools in the Klang Valley, an urban region in Malaysia, with its centre in Kuala Lumpur. Excel software will be used for the random selection and allocation of 4 out of these 25 Arabic schools. The first two schools are assigned intervention groups, while the next two are assigned control groups. Subsequently, 4 classes will be

| | STUDY PERIOD | | | | |
| --- | --- | --- | --- | --- | --- |
| | Enrolment | Allocation | Post-allocation | | |
| TIMEPOINT** | -$t_1$ | 0 | Baseline (T1) | One month post-intervention (T2) | 2months follow-up (T3) |
| **ENROLMENT:** | | | | | |
| Eligibility screen | X | | | | |
| Informed consent | X | | | | |
| Allocation | | X | | | |
| **INTERVENTIONS:** | | | | | |
| Intervention group: [Educational intervention program on preventing obesity] | | | ←——————→ | | |
| Control group: [ No Educational intervention program on preventing obesity] | | | ←——————→ | | |
| **ASSESSMENTS:** | | | | | |
| Sociodemographic Data | X | | | | |
| Knowledge, attitude, and practice of nutrition and physical activity | | | X | X | X |
| Physical activity [adapted questionnaire] | | | X | X | X |
| Sedentary activity [adapted questionnaire] | | | X | X | X |
| Food consumption frequency [adapted questionnaire] | | | X | X | X |
| Eating attitudes [adapted questionnaire] | | | | | |
| Health beliefs [adapted questionnaire] | | | | | |
| Body mass index (BMI) [anthropometric measurements] | | | | | |

**Fig 1. SPIRIT schedule of enrolment, interventions, and assessments.**

randomly selected from each school. Two of these classes will consist of 13-year-old seventh-grade students, while the next two will consist of mostly 14-year-old eighth-grade students.

## Study participants

The study participants will comprise Middle Eastern students in early adolescence, within the ages of 13 and 14 years, located in Arabic schools in the Klang Valley area. The inclusion criteria include Arabic students aged 13 to 14 years, with at least one parent consenting to their participation in this study. Students afflicted with illnesses such as asthma, diabetes, cancer, cardiovascular diseases, fractures, cirrhosis, or other diseases, as well as students exempted from physical activities due a medical condition, will be excluded from this investigation. The sampling unit will be represented by a randomly selected Arabic school Middle Eastern adolescent, who meets the inclusion and exclusion criteria set for this study.

## Ethical approval and informed consent

Ethical approval for the RCT was obtained from the Ethics Committee for Human Study of Universiti Malaya (UM) [UM.TNC2/UMREC_2039], and the directors of the selected schools. Written consent will be acquired from all parents, or the legal guardian, of the participants. Also, the participants, and their parents, will be privy to a description of the study, through an information sheet. The information provided covers: the purpose and procedures of the study, the unrestricted right of a participant to withdraw from the study, and the data collection process at three time periods (baseline, post-line, and two months after post-line). The participants and their parents, will be given the opportunity to inquire about the research, before extending their consent. Participants will be assured, that their participation in the study is entirely voluntary, and that they have the unrestricted right, to withdraw from the study at any time.

## Sample size

The cluster randomized control trial formula was applied to calculate the sample size, taking into consideration a power of 80%, and a 95% confidence level [28]. According to the proportion of overweightness among adolescents, calculated in a previous study conducted by James et al. [29]:

$$\text{n} = DE \times (Z_{1-\alpha/2} + Z_{1-\beta}) \frac{2P1(1-P1) + P2(1-P2)}{(P1-P2)2}$$

$$\text{DE} = 1 + p \text{ (m-1)}$$

**where**:

- $Z_{1-\alpha/2}$ = Standard score for (1–0.025) = 1.96

- $Z_{1-\beta}$ = Standard score (1–0.20) = 0.84 (80% power)

- P1 = The proportion of those overweight from control group (p1) = 27.6%

- P2 = The proportion of those overweight from Intervention group (p2) = 23.1%

- Intra-cluster correlation ($P$) = 0.05

- Number of individuals in each cluster (m)

- DE = Design Effect = 2.2

$$n = 2.2 \times (1.96 + 0.84) \frac{20.276(1 - 0.276) + 0.231(1 - 0.23)}{(0.276 - 0.23)2}$$

Sample size = 220
Adding 15% to the number of subjects as attrition rate = 250
Therefore, the total sample size is 250 participants (125 participants for each group).

## Sampling method

Simple random sampling will be used for the selection of participants. Two schools will be assigned the intervention group, and two schools the control group, followed by the selection of classes. With this form of probability sampling, researchers use a sampling frame, which includes a list of all students from the selected schools. For this study, a table of random numbers, will be used for the selection of participants, up to the point the sample size is attained. The random selection of participants is achieved by way of Excel software.

## Data collection

**Cluster randomized trial flow.** *Recruitment of participants*. The teachers and principals will be updated on the study objectives, and the student eligibility criteria, for the cluster randomised controlled trial (RCT). The selection of participants will be conducted at the selected schools. Only those who are available, during the accumulation of data, will be further briefed and assessed on their eligibility, according to the inclusion and exclusion criteria, stated in a screening form. The school administration will forward an invitation, to the parents of eligible participants, to obtain written consent, before the commencement of the study. On receipt of the written consent from students and their parents, the collection of baseline data will begin. The participant information sheet, in English and Arabic, will be made available to eligible students and their parents. The participants will be provided with an explanation, on the objectives and protocols of the study. All participants involved in this study are volunteers, and guaranteed the unrestricted right, to pull out of this study, at any time. Participants will be assured, that all information related to their identities, will be kept confidential. The random selection and allocation of participants, to the intervention or control groups, will be realized through the Excel software. Subsequently, the intervention group will be subjected to the educational intervention. For the post-intervention and follow-up assessments, the participants will be required to fill in the same questionnaire, albeit, excluding the section referring to personal information.

*Study principles*. This investigation is based on the study protocol item, recommendations for interventional trials (SPIRIT), 2013, statement [23], while complying with the consolidated standards of reporting trials (CONSORTs), statement [30]. The flowchart for the CONSORTs is displayed in Fig 2. CONSORT was developed by a group of researchers not only to identify problems associated with RCTs, but also to present the results of the research in a clear and concise manner, facilitating the reading and assessment of RCTs [30]. The CONSORT 2010 guideline were chosen because it is intended to improve the reporting of parallel- group randomized controlled trials. It enables better understanding of a trial design, conduct, analysis, interpretation and assessment of the validity of its results.

*Randomization*. Randomization is ensured throughout the study, using sealed opaque envelopes. An independent researcher places envelopes enclosing the names of the schools in a box, and randomly selects four envelopes to represent the intervention and control schools.

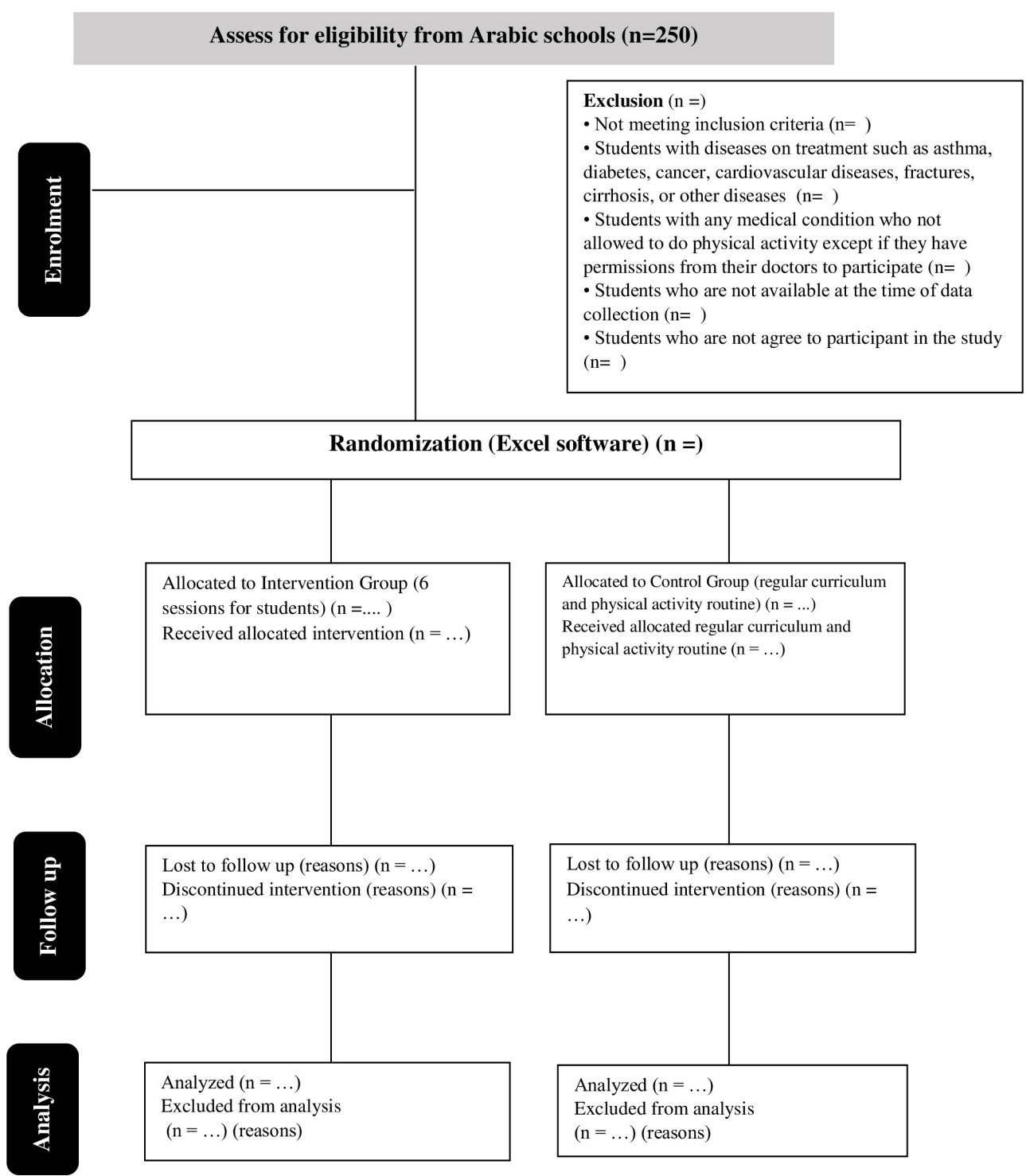

**Fig 2. Consort 2010 flow diagram.**

The independent researcher uses Excel software, for the selection of the participants. For this selection process, the IDs of all students will be entered into the datasheet, a random column will be created, and the formula " = RAND()" will be applied. Respondents will be assigned a

unique code, for the allocation of a concealment mechanism. This is to circumvent individual-level bias.

*Intervention*. 'Healthy lifestyle' intervention is based on the HBM theory, which considers the attitudes and beliefs of adolescents, for the development of their skills and favourable behaviours. A team of health promotion experts, from the University of Malaya's department of social and preventive medicine, with a special interest in obesity-related behaviour alteration, will assist the researcher, during the designing of the educational health promotion program.

The intervention is specifically designed to:

- Provide adolescents with the facts, statistics, and harmful effects, linked to obesity and overweightness.

- Encourage the consumption of healthy diets, to keep the risks of obesity at bay.

- Promote the indulgence in physical activity, to reduce the risks of obesity.

- Inspire students to adopt positive health behaviours.

- Develop a good decision-making and self-care attitude.

- Develop the three health belief elements of perceived susceptibility, perceived threat, and perceived benefit vs. barriers.

The improvement in knowledge and perceived disease risk, achieved through the educational programme, will lead to a better lifestyle, in terms of a healthier eating habit, and increased physical activity. It is anticipated, that the educational programme, will serve to promote the choice of healthy food, and the engagement in regular physical activity, among adolescents in their daily lives. The intervention will be carried out in a single-blind trial. While the researchers are knowledgeable regarding the details of the intervention, the adolescents will be kept in the dark regarding the existence of the control group, and the fact that they are about to be re-evaluated. The intervention process will include the distribution of educational booklets, as well as the running of educational classes.

The educational booklets, which will be distributed to participants, in the first session, are to be taken home and read, within a period of one week. The contents of the booklets include information on the risk factors associated to NCD, and the benefits to be gained from a healthy lifestyle and appropriate physical activity.

The educational classes, which include six weekly teaching units, are aimed at curbing obesity among the students, by equipping them with relevant knowledge, and behavioural skills. The separation of students, into groups of ten, is followed by the presentation of the training units by the researcher. For each 45-minute session, 15 minutes is set aside for the presentation of the topic through PowerPoint slides, 15 minutes for group discussions, and 15 minutes for group activities. The focus of the first and second training units, will be on educating the students, regarding the perils and complications, associated to being overweight or obese. The emphasis of the first session activity, will be on the measuring of the BMI, while the emphasis of the second session activity, will be on comprehending the nutritional information provided on food labels. The third teaching unit will emphasise on the selection of a healthy diet, which includes an appropriate type and amount of food. The third session activity will entail the preparation of a healthy serving of oatmeal. The fourth teaching unit will focus on education related to physical activities, including the appropriate type, duration, frequency, and intensity of physical activity. The fourth session activity will entail a 15-minute walk, within the school playground. The fifth teaching unit involves a discussion, on the benefits and barriers

**Table 1. Overview of the educational intervention classes to preventing obesity.**

| Sessions | Topics | Area of target | Intervention |
|---|---|---|---|
| Overweight and obesity | The dangers and complications of overweight and obesity | Knowledge of overweight and obesity | PowerPoint presentation, booklet, small group discussion |
| Reading the nutritional fact on the food label | To learn how to read and interpret food labels | What is a serving size and serving per container | PowerPoint presentation, booklet, Various packaged foods with food labels, Videos |
| Healthy diet | To understand food based approaches to reduce or prevent overweight and obesity | Knowledge of type and the amount of food<br>Choosing the healthy snack from the basket (decision-making). | PowerPoint presentation, booklet |
| Basics of physical activity | The physical activity types and recommended duration.<br>Benefits of physical activity<br>Physical activity opportunities. | To understand the importance of regular physical activity | PowerPoint presentation, booklet, Videos and workshops |
| Benefits and barriers of a healthy diet and physical activity program | Benefits of diet and physical activity<br>Barriers to adherence diet and physical activity participation | Belief in diet and physical activity adherence | PowerPoint presentation, booklet, Videos and workshops |
| Proposed methods to decrease Perceived barriers, and Enhance confidence and Decision-making skill. | To decrease perceived barriers and enhance confidence in adherence and decision-making skill | To motivate and increase an individual's confidence to take action to improve their healthy lifestyle | PowerPoint presentation, booklet, small group discussion |

associated to a healthy diet, and a fitness-promoting physical activity program. The session activity comes in the form of a role-playing exercise. For the sixth training unit, methods will be proposed for reducing the perceived barriers, enhancing adherence confidence, and improving of decision-making skills. The session activity will involve the selection of a healthy snack from a basket (decision-making skill). It is anticipated that with the proposed intervention, the participants will develop the practice of self-care, in the form of daily exercise and a healthy diet. This will go a long way towards the promotion of a healthy lifestyle, and the deterrence of obesity, among adolescents (Table 1).

Participants will receive motivational messages via WhatsApp during the follow-up. The messages in the form of short videos, pictures and flyers will be designed to deliver information on the risk factors associated to NCD, and the benefits to be gained from a healthy lifestyle and appropriate physical activity.

## Study outcomes

**Primary outcome.** The knowledge, attitude, and practice of nutrition and physical activity questionnaire (KAP-Q) will be used, to measure the primary outcome of interest for this study, in accordance with the approach employed by Sharif Ishak et al. [31] and Harake et al. [32]. The questionnaire consists of 73 items: knowledge (30), attitude (22), and practice (21). The knowledge section serves to determine the participants' level of comprehension, regarding nutrition and physical activity. The response options to the multiple-choice questions in the questionnaire are 'True', 'False' and 'I do not know'. Correct responses will be designated '1', while incorrect and 'I do not know' responses will be designated '0'. The maximum score is 30, and the minimum 0. A score above the mean is an indication of good knowledge. The five Likert scale responses for assessing attitude are 'strongly agree', 'agree' 'neutral', 'disagree', and 'strongly disagree'. Eleven negatively termed questions will be inversely recoded during data analysis. The maximum score is 110, and the minimum 22. A higher score is an indication of a greater positive attitude. The practice section includes 21 questions, which come with five Likert scale responses: 'very frequently', 'often', 'sometimes', 'rarely', and 'never'. Nine

negatively termed questions will be inversely recoded during data analysis. The maximum score is 105, and the minimum 21. A higher score is an indication of a more positive practice.

**Secondary outcome.** The secondary outcome variables in this study are:

*Change in the adolescent physical activity behaviour.* The physical activity questionnaire, for older children (PAQ-C), will be used to gauge the physical activity level of adolescents. The PAQ-C was applied for this study as it is deemed appropriate for schoolchildren aged between 8 and 14 years. This physical activity questionnaire, developed and validated by Kowalski [33, 34], is a self-supervised, 7-day questionnaire, that calculates moderate to healthy physical activity levels, during the school year [33, 34]. The personal administration of the questionnaire will be performed in a classroom. The PAQ-C holds ten items, with the score for each item ranging from 1 (low physical activity) to 5 (high physical activity), and the mean score for all items constitutes the overall PAQ-C score. To illustrate, item number 1 (rating 1 for 'no' and 5 for '7 times or more') on the activity checklist, forms a composite score for item 1. Items from 2 to 8 (score 1 for the lowest, and 5 for the highest activity response) mean. It is essential that for item 9, the mean for all days of the week is taken into consideration, for a composite score (1 for 'none' and 5 for 'very often'). Item 10 (PAQ-C) will not be contributed towards the final physical activity score. Its purpose is to identify students, who were unable to perform regular physical work during the week, due to illness or any other reasons. The mean score for the nine items constitutes the final PAQ-C activity summary score [34, 35]. According to Cronbach's alpha r = 0.777 [36], the reliability of PAQ-C, in terms of physical activity level identification, is 0.777. A total mean score between 1 and 2.33 indicates a low level of physical activity, a mean score between 2.34 and 3.66 indicates a moderate level of physical activity, while a mean score between 3.67 and 5 indicates a high level of physical activity [37].

*Change in adolescent sedentary activity.* The adolescent sedentary activity questionnaire (ASAQ) will be used to determine sedentary time. The questionnaire queries participants, on the amount of time they spend in eleven different sedentary behaviours per day, during weekdays and during weekends. The questionnaire separates the sedentary behaviours into five categories: screen time (watching television, watching videos and DVDs, using a computer for entertainment, using an iPad or smartphone for entertainment, as well as playing video games such as Xbox, Wii, Nintendo, and PlayStation), education, (using the computer for studying, studying without using the computer and being tutored), travel (by car, bus or train), cultural activities (handicraft pastimes, hobbies or reading for recreation), social activities (chatting with friends, using a phone, chilling, and attending prayer sessions at the mosque). The total time for each activity of each category is first determined, followed by the calculation for total time of all categories, to determine the total sedentary time per weekday and per weekend [38]. A total time of (ASAQ) ≥ 4 hours/day is an indication of high sedentary behaviour, while a total time of < 4 hours/day is an indication of low sedentary behaviour [39]. The test-retest correlations are ≥ 0.70 [38].

*Change in food consumption frequency.* Semi-quantitative food-frequency questionnaire (SFFQ) will be used, to measure frequently consumed food items, in terms of time and total calories [40]. The SFFQ names 74 food items, and the categories for intake frequency are once per day, 2–3 times per day, 4–5 times per day, 6–7 times per day, 1–2 times per week, 3–4 times per week, 5–6 times per week, once a month, 2–3 times monthly, or never. The 74 items of nutrients, and the reproducibility rates of the food groups, are 0.46 and 0.49 respectively. Participants are queried on the type of food they consume, as well as the frequency of consumption and quantity [41].

*Change in the eating attitudes.* The eating attitudes test-26 (EAT-26) will be used, to collect data on the prevalence of eating disorders [42]. The eating attitude test is a 26-item questionnaire, widely applied to measure the symptoms and concerns, related to eating disorders in

adults and adolescents, aged 13 years and above. The three main subscales of EAT-26 are dieting (13 items), bulimia as well as food preoccupation (6 items), and oral control (7 items). The dieting subscale questions include those associated to overweight anxiety, the desire to lose weight, vomiting, and the urge to vomit. The bulimia and food preoccupation subscale questions cover issues, associated to the persistent fixation over food. And the oral control subscale questions focus on the pressure, exerted by people in general, to be weight-conscious. Items are rated on a 6-point Likert scale ranging from 'always' (6), to 'never' (1). The six-point classification for the items from 1 to 25 extend from 'always' to 'never'. A score of 3 is assigned for 'always', 2 for 'usually', 1 for 'often', and 0 for 'sometimes', 'rarely' and 'never'. As for the 26[th] item, a score of 0 is assigned for 'always', 'usually' and 'often', while a score of 1 is assigned for 'sometimes', 2 for 'rarely' and 3 for 'never'. A score of 20 and above is indicative of an eating disorder [43]. The reliability of (EAT-26) is rated 0.87 by Cronbach's α [44].

*Change in health beliefs*. Health belief will be measured by way of a self-report health belief questionnaire (HBQ). The breakdown of the 89 questions, representing eight perception and behavioural categories, are as follows: 13 questions on severity perception, separated into three subscales (emotional/mental health, physical health/fitness, and social/professional); 7 questions on susceptibility perception, separated into two subscales (lifestyle and environment); 14 questions on perceived barriers, separated into three subscales (practical concerns, emotional/mental health, and awareness); 13 questions on perceived benefits, separated into three subscales (emotional/ mental health, physical health/fitness, and social/professional); 12 questions on action cues, separated into two subscales (internal and exterior action cues); 18 questions on dietary self-efficacy, separated into two subscales (habits and preferences, and emotional/mental health); 7 questions on exercise self-efficacy, and five questions on weight management behavioural intention, separated into two subscales (dieting and exercising). All statements are rated by way of a five-point Likert scale, ranging from 1 (strongly disagree), to 5 (strongly agree). A higher score indicates a greater level of belief [45].

*Change in body mass index (BMI)*. The body mass index (BMI) will be reported in kg/m^2, by way of the formula BMI = kg/m2, in which kg is a person's weight in kilograms, and m2 is his/her height in metres squared. Body weight will be measured using Omron HBF 375, to the nearest 0.1 kg. SECA body meter 206 will be employed to measure height, to the nearest 0.1 cm. Body mass index (BMI) will be classified into underweight, normal weight, overweight, and obese). A BMI less than 18.5 falls within the underweight range, a BMI of 18.5 to <25 falls within the healthy weight range, a BMI of 25.0 to <30 falls within the overweight range, and a BMI of 30.0 and above falls within the obese range. Additionally, the BMI-for-age (z-score) will be analysed using the WHO AnthroPlus version 1.0.4 software, while the WHO Growth Reference will be applied to categorise body weight status, using the reference data for children and adolescents aged 5–19 years.

**Questionnaire and study module translation.** The forward and back translation processes will be utilised, to translate the study's English-language questions. All queries are initially translated from English into Arabic. Subsequently, a professional linguist will translate the original questionnaire, from English to Arabic, in the forward direction (native speaker of Arabic and expert in English). A separate impartial, knowledgeable, and skilled translator will then perform the reverse translation from Arabic to English (native speaker of Arabic and expert in English). To remove the possibility of any discrepancy between the items in the original and translated versions, another English language translator is brought into the picture, to compare the back-translated English version with the original questionnaire. Ultimately, the accuracy, of the final version of the module and questionnaire, is verified by the researcher, through a comparison between the translated version, and the original text.

## Statistical analysis

Data analysis will be conducted using the statistical package for social sciences (SPSS) software version 26 for Microsoft Windows (Chicago, IL, USA). An inspection for omitted values and outliers will be conducted prior to data analysis. Descriptive statistics will be applied for the presentation of the data to determine means, median, standard deviations (continuous variables), frequencies, and percentages, to represent categorical data (categorical variables). The normality of variables will be described among continuous data by way of the Kolmogorov-Smirnov test, as well as the Skewness & Kurtosis test. In the Kolmogorov- Smirnov test, a P-value > 0.05 is an indication of normal data distribution. Data not normally distributed, will be transformed into a categorical variable. In the Skewness & Kurtosis test, -1.96 to +1.96 is deemed normal data.

One Way ANOVA will be used, to determine whether any significant within-group variances occurred over time, for the selected variables, if the data is normally distributed, while Friedman's test will be used, to determine whether any significant within-group variances occurred over time, for the selected variables, if the data is not normally distributed.

The generalized estimating equation (GEE) will be used, to test the effect of the nutrition intervention program, on the selected variables (outcomes), between & within the group, at baseline, six weeks, and two months following intervention, which is adjusted for clustering. Missing data will not be replaced. Outcomes will be assessed at each time point, along with derived average overall three-time points, thus ensuring the delivery of both the cumulative and overall effects.

## Potential confounders

The socio-demographic variables, which will be identified as potential confounders, include age (years), gender (male or female), country of origin, ethnicity, highest education level for parents, parents' occupation and socioeconomic status of the family.

## Results

The rules for the consolidated standard of reporting trials (CONSORT) will be complied with during the reporting of the study results. The results derived through this research, which will be published in a global open-access, peer-reviewed publication, will also be communicated to educational and medical establishments, as well as national and international conferences, by the lead investigator.

The datasets will be used and/or analysed during the current study, and will accessible through the corresponding author, upon request.

## Discussion

This study protocol, delves into the use of the randomized controlled trial (RCT) design, to examine the effects of an intervention program, on the physical activity and diet behaviour, as well as on the behavioural intention of weight management, perceived self-efficacy with regards to exercise, perceived self-efficacy with regards to dieting, cue to action, perceived benefits, perceived barriers, perceived susceptibility, and perceived severity, of Middle Eastern adolescents in Arabic schools, located in Malaysia. It is anticipated, that with the implementation of this integrated intervention (titled Healthy Lifestyle), the health status of Middle Eastern adolescents in Malaysia, will be enhanced. The effectiveness of this intervention programme will be is reinforced by its adaption (through field assessments) to coincide with the Malaysian setting.

Activities in the intervention programme will be formulated for integration, into the students' routine practices. The RCT can serve to generate information, on the benefits to be gained from regular physical activity and a healthy diet, as well as reduce the barriers to non-adherence. Non-adherence represents the main challenge to obesity control globally.

The maintenance of a healthy weight is crucial for reducing not only the risk of heart disease, strokes, diabetes, and high blood pressure, but also several types of cancer.

This study will contribute towards the existing body of knowledge in this area, thus playing a role in promoting a healthy lifestyle among adolescents. The findings from this study can also assist health policymakers, during their efforts to develop effective strategies, for the prevention of obesity and overweightness, among Arabic adolescents in Malaysia, particularly students. Additionally, the information gathered from this undertaking, will provide a better understanding, of the unhealthy habits practiced by Arabic adolescents residing in Malaysia. This information, can contribute towards the development of a more comprehensive programme for the promotion of a healthy lifestyle among Middle Eastern adolescents.

The student population can take advantage of the opportunities, and activities, provided by this programme, to increase their awareness, regarding the consequences associated to negative health-related behaviours.

To summarize, this RCT will prove to be useful, during investigations on the impact of physical activity, and healthy diet behaviours, on educational intervention. According to our study hypothesis, the intervention programme is feasible, and can facilitate the development of knowledge, attitudes and practices, with regards to body weight status, physical activity level, nutrition status (BMI and dietary intake), and disordered eating, among Middle Eastern adolescents, in Arabic schools located in Malaysia. This intervention can be adjusted for application to Arab adolescents in general, and to other age groups, including adults and the elderly. This study presents an effective approach, for evaluating intervention programmes, focusing on the physical activity level and eating behaviour, among Middle Eastern adolescents in Malaysia. Besides the educational modules and materials, this study also offers references and teaching materials, which are accessible to school authorities and parents. In terms of limitations, the scope of this investigation is confined to schools, located in a specific region of Malaysia.

## Supporting information

**S1 Checklist.**
(DOC)

**S1 File.**
(PDF)

## Author Contributions

**Conceptualization:** Hanan Al-Haroni, Nik Daliana Nik Farid.

**Data curation:** Hanan Al-Haroni, Nik Daliana Nik Farid.

**Methodology:** Hanan Al-Haroni.

**Project administration:** Nik Daliana Nik Farid, Mohamad Shafiq Azanan.

**Software:** Hanan Al-Haroni.

**Supervision:** Nik Daliana Nik Farid, Mohamad Shafiq Azanan.

**Validation:** Hanan Al-Haroni.

**Visualization:** Mohamad Shafiq Azanan.

**Writing – original draft:** Hanan Al-Haroni.

**Writing – review & editing:** Hanan Al-Haroni, Nik Daliana Nik Farid, Mohamad Shafiq Azanan.

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
