## [Decision Letter · Decision Letter 0]

24 Apr 2023

PONE-D-23-05657Effectiveness of education intervention, with regards to physical activity level and a healthy diet, among Middle Eastern adolescents in Malaysia: a study protocol for a randomized control trial, based on a health belief modelPLOS ONE

Dear Dr. Al-Haroni,

Thank you for submitting your manuscript to PLOS ONE. After careful consideration, we feel that it has merit but does not fully meet PLOS ONE’s publication criteria as it currently stands. Therefore, we invite you to submit a revised version of the manuscript that addresses the points raised during the review process.

A

The manuscript discuss very important topic “ obesity among adolescent”. Your paper needs to revised to meet the following comments:

You need to extended more details about CONSORT. See: Consort 2010 statement: extension to cluster randomised trials.Do the authors have access to further baseline data such as, e.g highest education level for parents, or ethnicity, as this might also play a role in confounding with outcome.As the program was 6 weeks, was compliance measured in any way?There are more similar studies therefore this is not the first intervention study. So the claim is false and should be changed after review.It is good that HBM model is being used. But there seems that only one model has been reviewed for developing intervention. The suggestion is that to read more models, look for advantages and disadvantages and then select. Even a multi-model strategy has been used for developing intervention. For example, WHO has suggested one TMT, BCW is another one. You need to review all the models and choose the model/s that are more appropriate in your context.Must read about issues and problems of using HBM before rejecting it or using it along with other models.

We look forward to receiving your revised manuscript.

Kind regards,

Fadwa Alhalaiqa

Academic Editor

PLOS ONE

3. We note that the original protocol that you have uploaded as a Supporting Information file contains an institutional logo. As this logo is likely copyrighted, we ask that you please remove it from this file and upload an updated version upon resubmission.

a) If there are ethical or legal restrictions on sharing a de-identified data set, please explain them in detail (e.g., data contain potentially sensitive information, data are owned by a third-party organization, etc.) and who has imposed them (e.g., an ethics committee). Please also provide contact information for a data access committee, ethics committee, or other institutional body to which data requests may be sent. Please note that authors, including Corresponding Authors, are not permitted to be the sole point of contact for data requests.

b) If there are no restrictions, please provide the minimal anonymized data set necessary to replicate your study findings as either Supporting Information files or to a stable, public repository and provide us with the relevant URLs, DOIs, or accession numbers. For a list of acceptable repositories, please see http://journals.plos.org/plosone/s/data-availability#loc-recommended-repositories.

Additional Editor Comments:

The manuscript discuss very important topic “ obesity among adolescent”. Your paper needs to revised to meet the following comments:

1. You need to extended more details about CONSORT. See: Consort 2010 statement: extension to cluster randomised trials.

2. Do the authors have access to further baseline data such as, e.g highest education level for parents, or ethnicity, as this might also play a role in confounding with outcome.

3. As the program was 6 weeks, was compliance measured in any way?

4. There are more similar studies therefore this is not the first intervention study. So the claim is false and should be changed after review.

5. It is good that HBM model is being used. But there seems that only one model has been reviewed for developing intervention. The suggestion is that to read more models, look for advantages and disadvantages and then select. Even a multi-model strategy has been used for developing intervention. For example, WHO has suggested one TMT, BCW is another one. You need to review all the models and choose the model/s that are more appropriate in your context.

6. Must read about issues and problems of using HBM before rejecting it or using it along with other models.

For further details see the reviewers feedback:

Reviewer 1: This is an interesting study addressing a very important worldwide concern re obesity in adolescents. This study delves into the effect of an Health belief Model -based education program, on the physical activity and eating behaviour, among Middle Eastern adolescents in Malaysia

-The authors have mentioned following the CONSORT. Can this be extended to look at the Research Methods & Reporting, Consort 2010 statement: extension to cluster randomised trials.

- In addition do the authors have access to further baseline data such as, e.g highest education level for parents, or ethnicity, as this might also play a role in confounding with outcome.

-As the program was 6 weeks, was compliance measured in any way?

Reviewer 2:

I appreciate the effort and the clarity with which the study has been designed. Methodology is presented nicely and is clear. To improve the protocol, I have following suggestions:

1. There are more similar studies therefore this is not the first intervention study. So the claim is false and should be changed after review.

2. It is good that HBM model is being used. But there seems that only one model has been reviewed for developing intervention. I would suggest that read more models, look for advantages and disadvantages and then select. Even a multi-model strategy has been used for developing intervention. For example, WHO has suggested one TMT, BCW is another one, but I would encourage you to review all the models and choose the model/s that are more appropriate in your context.

Must read about issues and problems of using HBM before rejecting it or using it along with other models.

Best wishes

Reviewers' comments:

Reviewer's Responses to Questions

**Comments to the Author**

1. Does the manuscript provide a valid rationale for the proposed study, with clearly identified and justified research questions?

Reviewer #1: Yes

Reviewer #2: Yes

2. Is the protocol technically sound and planned in a manner that will lead to a meaningful outcome and allow testing the stated hypotheses?

Reviewer #1: Yes

Reviewer #2: Yes

3. Is the methodology feasible and described in sufficient detail to allow the work to be replicable?

Reviewer #1: Yes

Reviewer #2: Yes

4. Have the authors described where all data underlying the findings will be made available when the study is complete?

Reviewer #1: No

Reviewer #2: Yes

5. Is the manuscript presented in an intelligible fashion and written in standard English?

Reviewer #1: Yes

Reviewer #2: Yes

6. Review Comments to the Author

You may also provide optional suggestions and comments to authors that they might find helpful in planning their study.

Reviewer #1: This is an interesting study addressing a very important worldwide concern re obesity in adolescents. This study delves into the effect of an Health belief Model -based education program, on the physical activity and eating behaviour, among Middle Eastern adolescents in Malaysia

the authors have mentioned following the CONSORT. Can this be extended to look at the Research Methods & Reporting

Consort 2010 statement: extension to cluster randomised trials.

In addition do the authors have access to further baseline data such as, e.g highest education level for parents, or ethnicity, as this might also play a role in confounding with outcome.

As the program was 6 weeks, was compliance measured in any way?

Reviewer #2: I appreciate the effort and the clarity with which the study has been designed. Methodology is presented nicely and is clear. To improve the protocol, I have following suggestions:

1. There are more similar studies therefore this is not the first intervention study. So the claim is false and should be changed after review.

2. It is good that HBM model is being used. But there seems that only one model has been reviewed for developing intervention. I would suggest that read more models, look for advantages and disadvantages and then select. Even a multi-model strategy has been used for developing intervention. For example, WHO has suggested one TMT, BCW is another one, but I would encourage you to review all the models and choose the model/s that are more appropriate in your context.

Must read about issues and problems of using HBM before rejecting it or using it along with other models.

Best wishes

7. PLOS authors have the option to publish the peer review history of their article (what does this mean?). If published, this will include your full peer review and any attached files.

Reviewer #1: No

Reviewer #2: No

---

## [Author Response · Author response to Decision Letter 0]

4 Jul 2023

We thank all reviewers for their valuable comments. We have modified the manuscript according to each of the respective reviewers. We hope that the reviewers and editors will satisfy with the amendments in the revised manuscript.

Details are listed in Response to Reviewers file.

---

## [Editor Report · Decision Letter 1]

17 Jul 2023

PONE-D-23-05657R1Effectiveness of education intervention, with regards to physical activity level and a healthy diet, among Middle Eastern adolescents in Malaysia: a study protocol for a randomized control trial, based on a health belief modelPLOS ONE

Dear Dr. Nik Daliana Nik Farid,

Thank you for submitting your manuscript to PLOS ONE. After careful consideration, we feel that it has merit but does not fully meet PLOS ONE’s publication criteria as it currently stands. Therefore, we invite you to submit a revised version of the manuscript that addresses the points raised during the review process.

The CONSORT  why you choose this guideline not others.?You need to answer the question: Do the authors have access to further baseline data such as, e.g highest education level for parents, or ethnicity, as this might also play a role in confounding with outcome? By saying whether the authors have access or not and what are these data.=========================== Please submit your revised manuscript by Aug 31 2023 11:59PM. If you will need more time than this to complete your revisions, please reply to this message or contact the journal office at plosone@plos.org. Please include the following items when submitting your revised manuscript:A rebuttal letter that responds to each point raised by the academic editor and reviewer(s). You should upload this letter as a separate file labeled 'Response to Reviewers'.A marked-up copy of your manuscript that highlights changes made to the original version. You should upload this as a separate file labeled 'Revised Manuscript with Track Changes'.An unmarked version of your revised paper without tracked changes. You should upload this as a separate file labeled 'Manuscript'.If applicable, we recommend that you deposit your laboratory protocols in protocols.io to enhance the reproducibility of your results. Protocols.io assigns your protocol its own identifier (DOI) so that it can be cited independently in the future. For instructions see: https://journals.plos.org/plosone/s/submission-guidelines#loc-laboratory-protocols. Additionally, PLOS ONE offers an option for publishing peer-reviewed Lab Protocol articles, which describe protocols hosted on protocols.io. Read more information on sharing protocols at https://plos.org/protocols?utm_medium=editorial-email&utm_source=authorletters&utm_campaign=protocols.

We look forward to receiving your revised manuscript.

Kind regards,

Fadwa Alhalaiqa

Academic Editor

PLOS ONE

Journal Requirements:

Additional Editor Comments:

Dear Dr Nik Daliana Nik Farid,

The paper is improved however, You need to :

1- The CONSORT why you choose this guideline not others.?

2- You need to answer the question: Do the authors have access to further baseline data such as, e.g highest education level for parents, or ethnicity, as this might also play a role in confounding with outcome? By saying whether the authors have access or not and what are these data.

3- . Please ensure that your manuscript meets PLOS ONE's style requirements, including those for file naming. The PLOS ONE style templates can be found at

---

## [Author Response · Author response to Decision Letter 1]

27 Jul 2023

We thank all reviewers for their valuable comments. We have modified the manuscript according to each of the respective reviewers. We hope that the reviewers and editors will satisfy with the amendments in the revised manuscript. Details are listed in Response to Reviewers file.

---

## [Editor Report · Decision Letter 2]

31 Jul 2023

Effectiveness of education intervention, with regards to physical activity level and a healthy diet, among Middle Eastern adolescents in Malaysia: a study protocol for a randomized control trial, based on a health belief model

PONE-D-23-05657R2

Dear Dr. Nik,

We’re pleased to inform you that your manuscript has been judged scientifically suitable for publication and will be formally accepted for publication once it meets all outstanding technical requirements.

Kind regards,

Fadwa Alhalaiqa

Academic Editor

PLOS ONE
---

## [Editor Report · Acceptance letter]

2 Aug 2023

PONE-D-23-05657R2 

Effectiveness of education intervention, with regards to physical activity level and a healthy diet, among Middle Eastern adolescents in Malaysia: a study protocol for a randomized control trial, based on a health belief model 

Dear Dr. Nik Farid:

I'm pleased to inform you that your manuscript has been deemed suitable for publication in PLOS ONE. Congratulations! Your manuscript is now with our production department. 

Kind regards, 

on behalf of

Pro Fadwa Alhalaiqa 

Academic Editor

PLOS ONE